# Impact of Paternal Leadership on Employee Retention during COVID-19: Financial Crunch or Financial Gain

**José Moleiro Martins** [1,2], **Uzma Kashif** [3,*], **Rui Miguel Dantas** [1], **Muhammad Rafiq** [3] **and João Luis Lucas** [4]

1 ISCAL (Instituto Superior de Contabilidade e Administração de Lisboa), Instituto Politécnico de Lisboa, Avenida Miguel Bombarda, 20, 1069-035 Lisboa, Portugal
2 Instituto Universitário de Lisboa (ISCTE-IUL), Business Research Unit (BRU-IUL), 1649-026 Lisboa, Portugal
3 Department of Management Sciences, Superior University, Lahore 42000, Pakistan
4 Departamento de Gestão, Universidade de Évora, Largo Dos Colegiais, 2, 7002-554 Évora, Portugal
* Correspondence: uzmanoor1979@gmail.com

**Abstract:** The leadership style that is most appropriate for the given circumstance will determine whether or not a leader is successful. It means what great leaders should do while working with a diverse workforce. They should be emotionally intelligent in order to understand their team members and modify their leadership style in order to achieve the best out of them. Employee engagement in the workplace is crucial for firms, but different factors can keep employees motivated. Work engagement activities, particularly those supported by the human resource department, have typically been observed as the primary factors that motivate employees. However, the COVID-19 pandemic has brought about a number of adjustments. The primary goal of this study is to examine how virtual human resource practices and paternal leadership affected employee retention in COVID-19, with the function of work engagement activities as a mediator. The data were gathered from 250 Portuguese Professors who were instructing undergraduate students using a survey instrument. Smart-PLS partial least squares structural equation modeling (PLS-SEM) was used to assess the study's hypotheses. It has been discovered that paternalistic leadership, also known as a resource provider with a benevolent attitude, has a direct impact on job performance and employee intention to leave the job, but during a pandemic, where the role of the government in supporting their nationals was not as significant in Portugal as it could be, people also faced their leaders of organizations as opportunists. Not all, but most paternal organizations took financial decisions to safeguard their business and were not people-oriented. Now the dignity of the paternal leader on the canvas of leadership is fading. This neo-normal approach will contribute to the literature on paternal leadership.

**Keywords:** employee retention; financial objectives; paternal leadership; pandemic; work engagement; virtual human resource activities

## 1. Introduction

One of the most significant and successful solutions to the opportunities and concerns brought by the global context is leadership (Dagiene et al. 2022). The economic structure, independent public institutions, political stability, long-term constructive welfare programs for the public, entrepreneurial initiative support mechanisms, technical education, continuous research and development, and government support are key reasons for choosing a leadership style while commencing business initiatives in any country (Fatimah and Syahrani 2022).

Islam et al. (2020) identified that paternal leadership has the capacity to motivate, inspire, and demand high performance from others based on the deeply held core values of firms. The six leadership subscales listed below are generally equivalent to the attributes or styles of paternal leadership: visionary, inspirational, self-sacrificing, honest, decisive, and performance-oriented (Islam et al. 2022). In contrast to transactional leadership, this set of leadership practices is most strongly associated with paternal leadership theory.

Paternal leaders' charismas are different from transactional leaders' charismas in several ways (Ahmad et al. 2021).

Paternal leadership requires compassion and kindness in addition to being helpful and sensitive (Chaudhary et al. 2021). It has two subscales for leadership: humility and humanitarian orientation (Islam et al. 2019). In particular, the moral or ethical school of leadership theory informs paternal leadership. Although it has been argued for thousands of years, leadership ethics only became a distinct academic field of applied ethics in the 1990s (Maqsoom et al. 2022). Most of the study that has already been under taken on paternalistic leadership has focused more on its benefits, including increased team cohesion, job happiness, organizational commitment, and success in both roles and outside of them. The negative aspects of paternalistic leadership, however, have managed to remain hidden (Stein 2022). The paternalistic leadership found in Turkish culture as well as any potential drawbacks, including perceived job discrimination and nepotism has been highlighted (Boothe and Watson 2022). Because it is believed that the authoritarian nature of paternalistic leadership will induce the leader to discriminate against his followers, this study focuses on the connection between paternalistic leadership and employee discrimination and nepotism (Xu et al. 2022). Those who do not comply with the leader's authority and requests will be handled differently because of the leader's ultimate control and authority over his subordinates and the expectation of unquestioning adherence from them (Boothe and Watson 2022). Employee disobedience may merely cause the boss to treat the subordinates differently. Second, nepotism—the practice of doing extra favors for family members—is common in family-owned businesses (Lok et al. 2022). That makes sense in part because families work to ensure company continuity among generations in order to ensure the expansion of their inheritance. Paternalistic leadership styles are also quite prevalent in family-owned businesses (Michael-Tsabari et al. 2022). The leader, who is typically a family member, maintains total control over all decisions and has little regard for anyone else. This association suggests that nepotism and paternalistic leadership are related (Colovic 2022).

A considerable proportion of research has been under taken by social exchange theorists in support of the idea that an organization's commitment to its employees may be measured (Elmes 2022). The level of dedication that employees exhibit to the company, in turn, will directly depend on the organization. Consider the connection between the employer and employee as one of a fair transaction (Ndidi et al. 2022), with the way in which an employer treats employees having a direct impact on their performance, attitude, and commitment to the organization (Nishii and Leroy 2022). This is a valuable framework for analyzing commitment behaviors (Wang et al. 2022).

Employee attitudes and behaviors, including performance, mirror their expectations and impressions of the organization's treatment of them (Ćwikła et al. 2022). Human resource practices are shown to be significantly correlated with employee perceptions and attitudes in their multilevel model relating to human resource practices and employee reactions (Basu 2022). Employee attitudes, and more specifically, employee commitments, were linked to the interaction of human resource practices and perceptions, according to expert studies (Zaid and Jaaron Forthcoming). High-involvement work practices may improve employee retention, according to researchers. However, the majority of analyses of commitment and retention come from the perspective of the employer; as a result, new and improved initiatives are consistently presented (Aybas et al. 2022). Investments in high-involvement work practices may thus promote a pleasant work climate that may result in decreased turnover (Mulugeta 2022). These programs are intended to have a favorable influence on employee retention and commitment. Organizations have prioritized reducing unneeded and undesirable employee turnover by implementing HR procedures and regulations (Mondejar and Asio 2022).

Any organization's HR function is incredibly important for employee retention. Workplace policy and procedure improvements, internal promotion opportunities, employee training, and bonus payments are a few examples of strategies for retaining employees

(Almaw 2022). During a restructuring, the HR department is in charge of conducting, recommending, and putting into practice employee retention measures (Yussif 2022). Work is crucial to a person's quality of life and is closely tied to it. Extending the concept of work beyond the limitations of the office, overall life happiness has been linked to job satisfaction (Yussif 2022). As a result, employment is more than just a means of subsistence for people; it also helps them discover "purpose, stability, and a feeling of community and identity" (Neumüller 2022). Furthermore, firms must quickly adapt to a dynamic environment and allow their employees to thrive in the workplace if they are to maintain competition and success (Akkaya et al. 2022). Many studies have shown how important both work-related and non-work-related help, such as family and social support, is in making people more committed to and good at their jobs (Piehl 2022).

As a result of the ongoing COVID-19 pandemic, the world is currently experiencing an unparalleled crisis. In the middle of the spring semester of 2020 (Xiong et al. 2021), faculty members in educational organizations—particularly higher education institutions—abandoned face-to-face instruction and quickly transitioned to online learning (McLean and Warren 2022). Teachers had to balance the demands of their students with their own personal safety, maintain connections, and ensure quality while, for some, concurrently caring for loved ones and keeping an eye on their own children's development in their online education. One of the many key uncertain and immediate challenges brought on by the pandemic that has impacted educational institutions, students, programs, instructors, staff, and those who lead these organizations is the shift to digital learning. Higher education institutions' leadership structures swiftly changed, with senior executives originally charged with making quick decisions while prioritizing the health and safety of students, teachers, and staff.

Any institutional crisis is defined by its triggering events, but the reputation and status of an institution are frequently more influenced by societal construction and opinions about how they are handled. Because the faculty is regarded as the foundation of the education sector, the study is focused on educational institutions, particularly the faculties of private sector educational institutions. The faculty efforts are primarily on an intellectual level, concentrating on the future of the country. A faculty would need to help students understand the importance of cultivating curiosity, imagination, resilience, and self-control; they would also need to help students understand the importance of respecting and appreciating the ideas, perspectives, and values of others; dealing with rejection and failure; and moving forward in the face of adversity.

In the critical pandemic period, the educational institution's faculty members put these previously indicated skills into practice. When they noticed for themselves the shifting demands and priorities of educational authorities during the COVID-19 outbreak, they too rapidly changed from being traditional teachers to e-instructors. This marked a turning moment where the world began to discern between actual leadership and coated leadership after the so-called leadership was exposed to the financial community. A pivotal moment that will be reflected in how many firms handle this crisis will be remembered for decades. Some educational institutions talk about having a social mission and set of values or about how much they appreciate their staff and other stakeholders, yet some firms continue to make money even when they have not lost a penny and do not anticipate doing so in the future. According to research, individuals only genuinely think that their company has a mission and guiding principles when they witness management choosing to place those principles higher than immediate financial success. The researcher is aware that executives were under pressure from investors and bankers to save money and lessen the likelihood of a loss, but neither the investors nor the bankers were willing to go hungry. It is important not only because it is the right thing to do as a business but also because it will cut down on the costs of rehiring staff, the difficulty of finding dedicated faculty members, and the loss of goodwill when the institute returns to normal operations.

The contribution of this study is two-fold; theoretically, the relationship between paternal leadership and employee retention with the mediating role of work engagement

during theCOVID-19pandemic was not explored in previous studies that were thoroughly investigated in this study, especially in the education sector. Moreover, a second major contribution of this study is to explore the relationships between the personality traits of paternal leaders with work engagement activities moderating the role of virtual human resource activities with the perspective of the pandemic in the education sector have not yet been explored before.

## 2. Literature

Particularly paternalistic CEOs can offer their staff resources or serve as a resource for them. Leaders are renowned for being kind but stern, which suggests that they frequently assist their followers, meet their needs, and challenge them to improve. Notably, both support and possibilities for progress can increase employee work engagement. Paternalistic leadership has been found in numerous studies to promote employee work engagement, work involvement, and dedication in Eastern cultures (Öge et al. 2018). In a sample of air traffic controllers, a direct correlation was discovered between paternalistic leadership and work engagement. This finding suggests that paternalistic leadership should be valued as a job resource, and employees should put more effort, involvement, and focus into their work in return. In the current research, the impact of five personality traits as independent variables is discussed (personality traits of paternal leader; influential, decisive, organized, empowering, and compassionate). The relationship between the personality traits of a paternal leader is discussed with the work engagement activities supported by virtual human resource practices. The study incorporated the virtual impact because of corona surge and abrupt stoppage of economic activities in the economies. The study incorporated the leadership perspective of employee retention. The objective of taking a leadership perspective in this research is important because the decision-making factors of a leader remain in discussion during and after theCOVID-19 period. Therefore, Section 2.1 explains the relationship between the personality traits of a paternal leader and work engagement. Section 2.2 explains the relationship among personality traits of a paternal leader, virtual human resource practices, and work engagement. Section 2.3 explains the relationship between work engagement activities and employee retention. Section 2.4 explains the relationship among paternal leader traits, work engagement activities, and employee retention (Tuan 2018).

### 2.1. Linking Personality Traits of Paternal Leader with Work Engagement Activities

Paternalism is typically understood to be a leadership approach that blends authoritarianism and fatherly generosity. Personalized regard and care are displayed by paternalistic leaders, who also exercise control and centralize decision-making. Paternalistic leaders satisfy the "dual criteria" of harmony and compliance so that the coexistence of benevolence and authority comes from the father figure, who is authoritative, demanding, and disciplinarian, as well as nurturing, loving, and dependable. In the workplace, leaders are a crucial component of the immediate social network on which followers rely to make decisions. This could help to explain why strong leaders are renowned for exciting their teams, intimately understanding their needs, and inspiring them to work hard toward a common objective. Superior communication abilities enable a leader to persuade followers to support them and stick with the group. On the other side, a leader can demonstrate their influence by using their vast knowledge. Subordinates admire leaders when they demonstrate their competence because they can see that they are knowledgeable about what they are doing, which encourages them to work for the company for an extended period of time (Chen et al. 2014; Nazir et al. 2020).

### 2.2. The Influencing Trait and Work Engagement

Focusing on both of these domains of influence and work involvement is crucial. Influential communicators pay attention to both their body language and the words they employ. The leader needs to start speaking in an expressive manner and adopt a firmer tone.

An effective leader does not mumble or appear uncertain, which helps their team members comprehend the long-term goals and the reason they should stick with the company. Along with strengthening their ability to influence others through communication, leaders may also strengthen their capacity to influence others through knowledge while aligning them with the organization's destiny (Liu et al. 2020). Hence, the following hypothesis is postulated.

**Hypotheses 1 (H1).** *The influential trait of paternalistic leadership has a positive effect on work engagement.*

### 2.3. The Empowerment Trait and Work Engagement

Gaining the most from subordinates through well-chosen and planned work engagement activities is the goal of paternalistic leadership (Gong et al. 2020). Similar to how a parent would want their children to succeed through those work engagement activities that help them succeed in the assigned task, the leader wants them to flourish and grow. Through their matching engagement activities, employees' empowerment is achieved. People are engaged in accordance with their potential and capacity according to the vision and qualities of a parental leader. The micromanagement of work engagement activities must be carefully balanced with total autonomy (Rafiq et al. 2021). While work engagement activities do not provide employees with significant authority in terms of making decisions and establishing processes, the leader's role also does not include undermining or challenging the employees' actions.

**Hypotheses 2 (H2).** *The empowerment trait of paternalistic leadership has a positive effect on work engagement.*

### 2.4. The Compassionate Trait and Work Engagement

A leader must have empathy for their followers in order to inspire loyalty. If a leader does not have empathy and compassion by understanding people and allocating the work according to their interests and potential, they will not be able to relate to what their subordinates are going through. Paternalistic leadership is about making employees feel comfortable and valued through well-managed work engagement activities.

According to scientific studies, compassion is something that can be learned (De Stasio et al. 2019). The Center for Investigating Healthy Minds at the University of Wisconsin–Madison showed that active compassion meditation encouraged higher altruistic behavior in the participants.

**Hypotheses 3 (H3).** *The compassionate trait of paternalistic leadership has a positive effect on work engagement.*

### 2.5. The Decisive Trait and Work Engagement

Decision-making is concentrated in the hands of the leader under paternalistic leadership. The capacity to make wise judgments through carefully planned engagement activities not only calls for knowledge and skill but also for decisiveness. It may seem simple to be able to act quickly and accept the consequences, but any leader will tell you that it is not that simple. The proverb "With great power comes tremendous responsibility" sums up the challenges of being a leader perfectly. Even though a leader may believe they are ready to make those tough decisions, indecision can readily set in when faced with two bad or two good options (Christian et al. 2011).

**Hypotheses 4 (H4).** *The decisive trait of paternalistic leadership has a positive effect on work engagement.*

### 2.6. The Organized Trait and Work Engagement

A paternalistic leader also needs to be well-organized. Given that the decisions, processes, and objectives demand the leaders' full attention, it's critical that he or she be able to

manage the many threads through thoughtfully created work engagement activities. Maintaining control over the business of the corporation by paying attention to work engagement activities will aid in building trust with the subordinates (Handa and Gulati 2014).

**Hypotheses 5 (H5).** *The organized trait of paternalistic leadership has a positive effect on work engagement.*

### 2.7. The Influencing Trait and Employee Retention

It is crucial for a leader to have influence over others. Leadership has advantages for both people and businesses, and in particular, paternal leadership has both direct and indirect effects on staff retention. A paternal leader boosts employees' productivity by attaining corporate objectives and putting in place a compensation plan to keep them on board. Paternalism lowers the likelihood of staff turnover and boosts retention (Sheridan 1992).

**Hypotheses 6 (H6).** *The influential trait of paternalistic leadership has a positive effect on employee retention.*

### 2.8. The Empowering Trait and Employee Retention

Focus on a few key elements in your work engagement activities if you want to improve your ability to empower others. Create a space where people can express themselves honestly and receive feedback. Encourage subordinates to seek self-improvement by offering rewards (Varekamp et al. 2006). By improving checks and balances and exploiting potential failures as learning opportunities, risks and failure can be reduced. Share information with your subordinates rather than keeping it all to yourself.

**Hypotheses 7 (H7).** *The empowerment trait of paternalistic leadership has a positive effect on employee retention.*

### 2.9. The Compassionate Trait and Employee Retention

In order to be better able to understand the feelings of their employees and build more trust with them, leaders can incorporate compassion meditation into their daily routines. Meditation on compassion is not difficult to accomplish (Potter 1969).

**Hypotheses 8 (H8).** *The compassionate trait of paternalistic leadership has a positive effect on employee retention.*

### 2.10. The Decisive Trait and Employee Retention

Paternalistic leadership places the majority of the decision-making in the hands of the leader. Although it is not simple, the path to greater decisiveness demands perseverance. Leaders must have specific objectives for everything they do. Leaders can make wise decisions if they are aware of what they really want to accomplish (Potter 1969).

**Hypotheses 9 (H9).** *The decisive trait of paternalistic leadership has a positive effect on work engagement.*

### 2.11. The Organized Trait and Employee Retention

A paternalistic leader also needs to have strong organizational skills. Instead of focusing on the big picture, which is obviously important, attempt to divide duties, objectives, and processes into more manageable chunks. One of the leadership philosophies that demand a lot from leaders is paternalistic leadership. A leader who can implement this management style effectively must exhibit traits such as influence, the capacity to encourage others, compassion, decisiveness, and strong organizational abilities (Cloutier et al. 2015; Potter 1969).

**Hypotheses 10 (H10).** *The organized trait of paternalistic leadership has a positive effect on employee retention.*

*2.12. Relationship among Paternal Leader Traits, Virtual Human Resource Practices and Work Engagement Activities*

HR should take the lead in the creation, assessment, and evaluation of proactive workplace policies and practices that assist in recruitment and retaining individuals with the skills and competencies required for growth and sustainability in order to build a culture of engagement. The HRM system is a crucial organizational factor that managers and staff use to communicate the organization's values, objectives, and policies. Examples of HR practices that clearly demonstrate how high performance is defined and the rewards that are associated with meeting pertinent goals or targets include thorough training programs, performance-based reward systems, and promotion criteria. The relationships between line management and employees should also include HRM (Wood et al. 2020). This is corroborated by data that show a positive relationship between HR systems with consistent HR practices and organizational success. Paternal leader studies highlight the significance of the role supervisors play in influencing workers' work attitudes and performance, and HR should take the lead in the creation, measurement, and evaluation of proactive workplace policies. Low-quality exchange relationships are characterized by strictly contractual exchanges and one-way, downward influence, whereas high-quality exchange relationships feature respect and mutual influence between supervisors and employees (Wilton 2022).

**Hypotheses 11 (H11).** *Virtual human resource practices moderate the positive relationship between personality traits of paternal leader influence (H1a), decisive (H1b), organized (H1c), empowerment (H1d), compassionate (H1e) and work engagement activities.*

*2.13. Relationship between Work Engagement Activities and Employee Retention*

HR should take the lead in the design, assessment, and evaluation of proactive workplace policies and practices in order to promote a culture of participation. The desire of employees to leave their jobs was negatively impacted by work engagement, according to the most recent expert studies. This finding revealed why people who are very involved in their work are less likely to leave the company, increasing their retention rate inside the company. Employees who are highly engaged display high levels of vigor, enthusiasm, pride in their work, involvement, and focus while performing their duties. Additionally, they would be more content and enthusiastic about their work, which might encourage them to keep working (Al-Hajri 2020).

**Hypotheses 12 (H12).** *Work engagement has positive relationship with employee retention.*

*2.14. Relationship among Paternal Leader Traits, Work Engagement Activities and Employee Retention*

There is plenty of work on staff retention that focuses on how crucial it is for all kinds of businesses. Employee retention through compensation plans is cheaper than the expense of losing workers. Leadership benefits both employees and companies equally, and in particular, a paternal leader has a direct and indirect impact on retaining employees. A paternal leader boosts employees' productivity by attaining the company's goals and putting in place a compensation plan to keep them on board. Paternal leadership lowers turnover intention and improves staff retention (Lee et al. 2019).

**Hypotheses 13 (H13).** *Work engagement mediates the positive relationships between personality traits of paternal leader influence (H1a), decisive (H1b), organized (H1c), empowerment (H1d), compassionate (H1e) of paternal leader and employee retention. See Figure 1.*

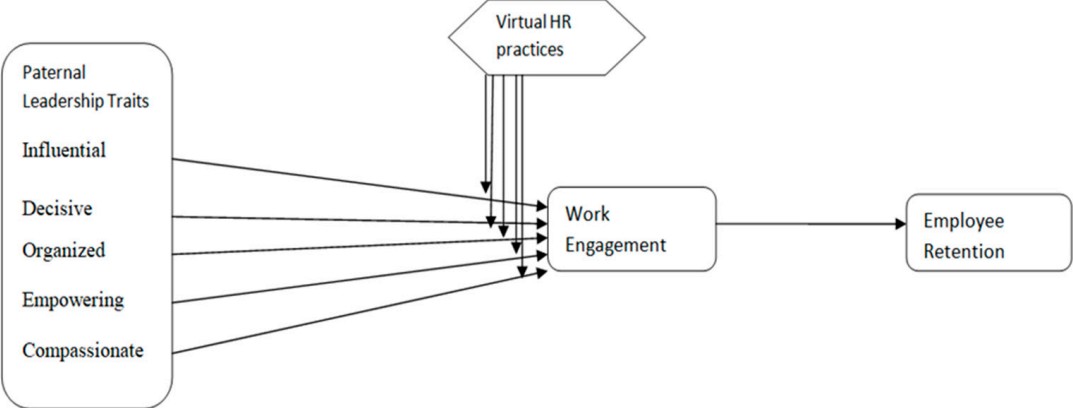

**Figure 1.** Theoretical Model.

## 3. Research Methodology

The methodology, the instrument, the data collection tool, and the sampling techniques used in the study are all included in this section. The research examines the connections between paternal leaders as resource providers, virtual human resource practices, work engagement, and employee retention. This study uses virtual human resource practices as a moderator variable in the context of Private Colleges in Lisbon.

In order to address the objectivity of the research issues, as recommended by experts, the study uses a positivist research paradigm. According to the ontological and epistemological stances of positivism, there is a singular reality and relationship between researcher and research. The study is distant and impartial. This study has adopted the positivistic paradigm in order to study the relationship between variables. In positivism, the researcher plays an unbiased role and uses a formal style of writing based on set principles (Park et al. 2020). The quantitative approach is regarded as the ideal approach for reaching the research objectives of the study because the current study is objective and practical in character. Additionally, the deductive method is the best method for achieving the generalization goal. As a result, the current research is transparent and supports positivism as the appropriate research approach to continue the process of data collecting and sampling.

### 3.1. Instrument

One of the most essential and crucial components of the study is the tool used to gather the data. For this study, a close-ended questionnaire (Appendix B) was used to collect the information. The study tool can be broken down into five key sections: Section 1 of the questionnaire covers the demographic information about the respondents, including their age, gender, job title, level of experience, etc. Section 2 elaborates on the personality traits of paternal leaders (Cui 2022; Gyamerah et al. 2022; Mitra and Roy 2021). (a). Influential (Bai and Pan 2022; Koritar 2022); (b). Decisive (Gunasekara et al. 2022; Nchey-Achukwu 2021); (c). Compassionate (Christensen and Learning 2022; Gilbert 2021; Reyes 2021); (d). Empowerment (Gyamerah et al. 2022; Kuznetsova 2022); (e). Organized (Lewis 2021; Montaudon-Tomas et al. 2022) Section 3. Virtual Human Resource Practices (Lee et al. 2022; Sposato and Rumens 2021), Section 4 explains the work engagement activities (Nal and Sevim 2021; Shafi et al. 2021; Tyagi 2021), and Section 5 explains employee retention (Gyamerah et al. 2022; Harvin 2022). The close-ended questionnaire was created with the research paradigm in mind to guarantee that the participants had unbiased responses. The survey uses a five-point Likert scale with a value range of strongly disagree to strongly agree, and a scale from one to five, where one is the least and five is the most. As a result of expert consultation and the findings of the pilot study, the questionnaire was contextually modified and adjusted. To ensure that the instrument was appropriate, a pilot research with 50 participants was conducted. Random sampling was used to identify the

sample for the pilot study. A pilot study was conducted to check the basic protocols of the research, particularly whether the instrument is appropriate or requires further work. On the basis of the pilot study, the researchers further continued the research process. The same participants were also used during the original data collection procedure. A random sample size strategy was applied to complete the pilot study.

*3.2. Sampling*

The top five colleges in Portugal were selected for this research. A purposive sampling approach was adopted to reach a relevant sample of the population from these colleges. The teachers were the participants in the study. Teachers who had a minimum of two years of experience were selected as respondents for this study. The sample size was calculated using Rao-soft software, which suggests the sufficiency of the respondents, as in Equations (1)–(3):

$$x = Z(c/100)2r\ (100 - r) \tag{1}$$

$$n = N\ x/((N - 1)E2 + x) \tag{2}$$

$$E = Sqrt\ [(N - n)\ x/n(N - 1)] \tag{3}$$

Twenty percent of the questionnaires were given to top management, while eighty percent were sent to middle management. Three hundred respondents in total filled out the questionnaire correctly and returned it. It took fifteen days for the respondents to fill out the questionnaire. The respondents' familiarity with the instrument's principles was validated; they were either working directly on systems with a similar nature or had the cognitive ability to comprehend and use the instrument.

## 4. Analysis

This research used Smart-PLS-3 software for structural modeling (SEM) in order to assess the measurement models and the structural equation modeling. Algorithms and bootstrapping are ways to assess the measurement model and structural model through structural equation modeling. The reason for using SMART-PLS was due to certain advantages over other software, such as the fact that it is effective for complex models, Smart-PLS also work better on small-size data, and there is no condition of normality. Smart-PLS is user-friendly and handy to use, due to which most researchers prefer to use it instead of other software.

*4.1. Descriptive Statistics of Respondents*

Table 1 shows the demographic statistics regarding the respondents, 47.6% of the respondents were male, and 52.6% were female. Overall, 50.4% of the respondents were younger than 30 years, 34.9% were between 31 and 40 years, 12.4% were between 41 and 50 years, and 2.3% of the respondents were above 50 years. The education level of the respondents represents 29.82% Master's degree employees, 30.08% of the respondents were M.Phil. employees, and 40.1% of the employees were doctors in their respective fields. In representing the employees' level in the organization, 12.4% were executive, 15.5% were senior leaders, 20.0% were senior managers, 23.3% were team leads, and 27.3% were non-team leads. The work experience of the employees represents 64.3% of employees who 'have less than 10 years' experience, 26.4% of the employees have between 10 and 20 years, and the rest (9.3%) have above 20 years.

**Table 1.** Descriptive Statistics of Respondents.

| Head | Description | Percentage |
|------|-------------|------------|
| Gender | Male | 36.4% |
| | Female | 63.6% |

**Table 1.** *Cont.*

| Head | Description | Percentage |
|------|-------------|------------|
| Age | Less than 30 | 50.4% |
| | 31–40 | 34.9% |
| | 41–50 | 12.4% |
| | Above 50 | 2.3% |
| Education Level | Masters | 29.82% |
| | M.Phil. | 30.08% |
| | Doctoral. | 40.1% |
| Levels in Organization | Executive | 12.4% |
| | Senior Leader\VP | 15.5% |
| | Senior Manager | 20.9% |
| | Team Lead | 23.3% |
| | Non-Team Lead | 27.9% |
| Work Experience | Less than 10 Years | 64.3% |
| | 10 to 20 Years | 26.4% |
| | Above 20 Years | 9.3% |

*4.2. Assessment of Measurement Model*

The content validity, discriminant validity, and convergent validity of the measurement model were all checked to confirm that they were valid. The objects were put through their paces to see if they accurately measured the variables. In order to ensure convergent reliability, statistics such as composite reliability, Average Variance Extracted (AVE), and factor loadings were considered. It was concluded that the factor loadings are the most important indicator of validity, and the value should be greater than 0.70. However, if the AVE values are not according to the threshold, then the researcher can delete the items up to twenty percent, and a minimum of a 0.50 threshold value for the outer leadings can be accepted. Items with values greater than 0.50 are kept in the database. Second, the composite reliability (CR) values are significantly higher than the cut-off value of 0.70, which is considered satisfactory. Third, acceptable AVE values are defined as those that are greater than 0.50 (Hair et al. 2019). The value of Cronbach's Alpha is greater than 0.70, which is sufficient to proceed with further analysis. This is the third sign of convergent validity. The details of the convergent validity are shown in Table 2.

**Table 2.** Reliability and convergent validity.

| | Cronbach's Alpha | rho_A | Composite Reliability | Average Variance Extracted (AVE) |
|---|---|---|---|---|
| Compassionate trait | 0.797 | 0.800 | 0.868 | 0.623 |
| Decisive Trait | 0.805 | 0.808 | 0.872 | 0.631 |
| Employee job retention | 0.892 | 0.895 | 0.918 | 0.653 |
| Empowerment trait | 0.884 | 0.889 | 0.911 | 0.595 |
| HRP & work engagement | 0.862 | 0.869 | 0.893 | 0.515 |
| Influential traits | 0.862 | 0.867 | 0.893 | 0.513 |
| Organized trait | 0.882 | 0.889 | 0.914 | 0.680 |

*4.3. Factor Loadings*

The table in Appendix A indicates the values of the outer factor loadings. The values show the contribution of the construct to the main variables. All of the values in the factor loadings must be greater than or equal to 0.60. The values depict that all of the values are greater than the threshold value of 0.60. The minimum value in the variable 'Alignment of Virtual Human Resource Practices with Work Engagement during COVID-19' is 0.611, and the maximum value is 0.860. Similarly, the maximum value of the 'Compassionate trait of Leader' is 0.836, and the minimum value is 0.748. Moreover, the values of the 'decision trait

of leader' fall between 0.834 and 0.754. Similarly, the values of 'employee job retention' are between 0.882 and 0.719. Furthermore, the values of the 'Empowerment Trait of Leader', 'Influential Trait of Leader', and 'Organized Trait of Leader' are also between 0.60 and 0.90. Two factors are deleted from the loadings due to their undesired contribution as we have a 20 percent margin to skip the factor. The items deleted were ctl2, dtl3, and etl7, and their values were less than the threshold value.

### 4.4. Discriminant Validity

The Fornel–Larcker criterion and the cross-loading assessment were the most commonly used in the evaluation of discriminant reliability. The Fornel–Larcker technique, on the other hand, did not accurately assess the discriminant validity (Janadari et al. 2018). For this reason, additional alternative approaches, such as the multitrait–multimethod matrix, were used in the current study to test discriminant validity. In addition, the heterotrait–monotrait correlation ratio was applied (Ab Hamid et al. 2017). Ringle et al. (2015) asserted that the HTMT, referred to as the heterotrait-to-monotrait ratio, must be smaller than 0.85 in order to demonstrate the discriminant validity in a test of discriminant validity. Because all of the values in Table 3 are less than 0.85, there is no concern with the discriminant validity. Discriminant reliability is important for determining the magnitude of the measurement inaccuracy. In order to identify whether the two concepts are linked or unrelated, it is necessary to adjust for the attenuation in the audio signal.

**Table 3.** Fornell–Larcker Criterion.

| | Compassionate Trait | Decisive Trait | Employee Job Retention | Empowerment Trait | HRP &Work Engagement | Influential Traits | Organized Trait |
|---|---|---|---|---|---|---|---|
| Compassionate trait | 0.789 | | | | | | |
| Decisive Trait | 0.633 | 0.794 | | | | | |
| Employee job retention | 0.625 | 0.642 | 0.808 | | | | |
| Empowerment trait | 0.781 | 0.725 | 0.741 | 0.771 | | | |
| HRP & work engagement | 0.696 | 0.667 | 0.809 | 0.745 | 0.717 | | |
| Influential traits | 0.677 | 0.768 | 0.636 | 0.745 | 0.656 | 0.716 | |
| Organized trait | 0.645 | 0.813 | 0.694 | 0.820 | 0.658 | 0.793 | 0.825 |

In addition to the Farnell and Larcker criterion, this study also used the HTMT ratio to identify the discriminant validity. According to Ringle et al. (2015), the values should be less than 1. Although some authors also suggest using values less than 0.90, this study fulfills the criteria of using less than 1. Table 4 indicates the values of HTMT.

**Table 4.** HTMT Ratios.

| Items | Compassionate Trait | Decisive Trait | Employee Job Retention | Empowerment Trait | HRP &Work Engagement | Influential Traits |
|---|---|---|---|---|---|---|
| **Decisive Trait** | 0.792 | | | | | |
| **Employee job retention** | 0.739 | 0.754 | | | | |
| **Empowerment trait** | 0.927 | 0.861 | 0.835 | | | |
| **HRP & work engagement** | 0.838 | 0.801 | 0.917 | 0.856 | | |
| **Influential traits** | 0.816 | 0.919 | 0.724 | 0.849 | 0.762 | |
| **Organized trait** | 0.769 | 0.963 | 0.778 | 0.922 | 0.747 | 0.903 |

### 4.5. Model Fit Indices

In Smart-PLS, the measure of fit is indicated by the values of SRMR, NFI, d-ULS, and chi-square. The values of SRMR should be less than 0.08. d-ULS can be indicated in inferential statistics for assessment. All of the values of the model fit indices are in the threshold values.

### 4.6. Regression Analysis

Path coefficients (β) and t-statistics were used to evaluate the structural model assessment (SEM) of the study (Table 5). The bootstrapping method was used to accomplish this goal, as previously stated. The hypotheses of the study are two-tailed, and a t-value of greater than 1.96 at a 5% level of significance is acceptable for each hypothesis, as suggested by (Hair et al. 2019). When it comes to the path coefficient -value, the higher the value, the greater the effect on the endogenous (dependent) variables is expected to be. The findings of the bootstrapping procedure are shown in detail in Table 6. Given that the 'Compassionate trait -> Employee job retention' has been demonstrated to be rejected ($p = 0.530$, t = 0.628), hence, H1 is considered rejected. The values of the second variable, 'Compassionate trait -> HRP & work engagement', are ($p = 0.015$, t = 2.435), which indicates that the compassionate trait is strongly associated with HRP and work engagement. The third variable, 'Decisive Trait -> Employee job retention', is not supported by the data. Furthermore, the third variable, which examines the relationship between the decisive trait and employee job retention, demonstrates a statistically insignificant relationship ($p = 0.874$, t = 0.159). Table 4 shows the results of this study. Moreover, 'Decisive Trait -> HRP & work engagement', is positively connected ($p = 0.030$, t = 2.172). Additionally, 'Empowerment trait -> Employee job retention'($p = 0.000$, t = 6.583) is the association between Empowerment trait -> Employee job retention. According to 'Influential traits -> Employee job retention' ($p = 0.966$, t = 0.043). The relationship between the influential traits and HRP & work engagement is also insignificant, as the values indicate ($p = 0.525$, t = 0.663). The values of the 'Organized trait -> Employee job retention' ($p = 0.168$, t = 1.382), so the contribution of the organized trait in employee job retention was found to have no relationship. Furthermore, the values of the 'Organized trait -> HRP & work engagement' depicts that there is no relationship between the organized trait and HRP& work engagement. Therefore, the hypothesis is rejected as per the values ($p = 0.663$, t = 0.432).

**Table 5.** Model Fit Indices.

|  | Saturated Model | Estimated Model |
|---|---|---|
| SRMR | 0.073 | 0.073 |
| d_ULS | 4.750 | 4.750 |
| d_G | 3.123 | 3.123 |
| Chi-Square | 1761.112 | 1761.112 |
| NFI | 0.622 | 0.622 |

**Table 6.** Mean, STDEV, t-Values, and *p*-Values.

| Items | Beta Values | STDEV | t-Values | *p*-Values | Hypothesis |
|---|---|---|---|---|---|
| Compassionate trait -> Employee job retention | −0.047 | 0.075 | 0.628 | 0.530 | Rejected |
| Compassionate trait -> HRP & work engagement | 0.235 | 0.096 | 2.435 | 0.015 | Accepted |
| Decisive Trait -> Employee job retention | −0.015 | 0.095 | 0.159 | 0.874 | Rejected |
| Decisive Trait -> HRP & work engagement | 0.218 | 0.100 | 2.172 | 0.030 | Accepted |
| Empowerment trait -> Employee job retention | 0.203 | 0.122 | 1.671 | 0.095 | Rejected |
| Empowerment trait -> HRP & work engagement | 0.383 | 0.124 | 3.095 | 0.002 | Accepted |

**Table 6.** *Cont.*

| Items | Beta Values | STDEV | t-Values | *p*-Values | Hypothesis |
|---|---|---|---|---|---|
| HRP & work engagement -> Employee job retention | 0.574 | 0.087 | 6.583 | 0.000 | Accepted |
| Influential traits -> Employee job retention | −0.005 | 0.108 | 0.043 | 0.966 | Rejected |
| Influential traits -> HRP & work engagement | 0.088 | 0.138 | 0.636 | 0.525 | Rejected |
| Organized trait -> Employee job retention | 0.197 | 0.143 | 1.382 | 0.168 | Rejected |
| Organized trait -> HRP & work engagement | −0.055 | 0.126 | 0.436 | 0.663 | Rejected |

## 5. Mediation Effect

Table 7 indicates the mediating effect of the variables. In this study, HRP and work engagement is the mediating variable between leadership traits and employee job retention. The existence of mediation explains the relationship between the other variables. The following tables indicate that the indirect effect is positive in the three variables; Compassionate trait -> HRP & work engagement -> Employee job retention, Decisive Trait -> HRP & work engagement -> Employee job retention and Empowerment trait -> HRP & work engagement -> Employee job retention, as their values are significant at level >0.05 it indicates that a mediation relationship exists. Additionally, the variables; Influential traits -> HRP & work engagement -> Employee job retention and Organized trait -> HRP & work engagement -> Employee job retention explains no mediation effect, as their values are above the significant values. (See Figure 2).

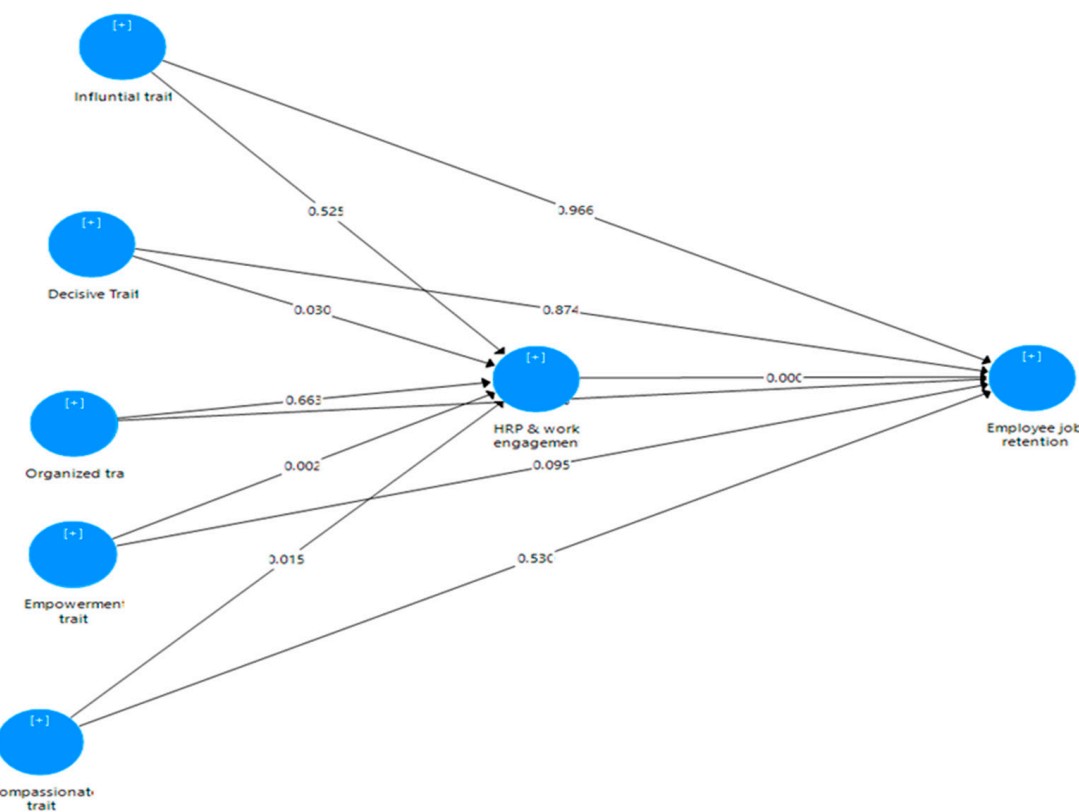

**Figure 2.** Mediation Effect.

**Table 7.** Mediation effect.

| | Original Sample (O) | Standard Deviation (STDEV) | t Statistics | *p*-Values |
|---|---|---|---|---|
| Compassionate trait -> HRP & work engagement -> Employee job retention | 0.135 | 0.062 | 2.185 | 0.029 |
| Decisive Trait -> HRP & work engagement -> Employee job retention | 0.125 | 0.062 | 2.025 | 0.043 |
| Empowerment trait -> HRP & work engagement -> Employee job retention | 0.220 | 0.079 | 2.794 | 0.005 |
| Influential traits -> HRP & work engagement -> Employee job retention | 0.050 | 0.082 | 0.614 | 0.539 |
| Organized trait -> HRP & work engagement -> Employee job retention | −0.032 | 0.075 | 0.424 | 0.672 |

## 6. Discussion

Benevolent and misleading leadership is exhibited by paternal leadership styles. A good-natured, paternalistic leader exhibits "parental affection" along with moderate authority in order to uphold social order in a peaceful setting. The exploitative paternal leader symbolizes a "darker" shade and appears to confirm a self-serving (or even dishonest) mentality of the leader, whose mild nurturing approach is utilized to ensure that followers are obedient and deferential. The economic system and both types of paternal leaders emphasize group loyalty over individual goals, which is more in line with the sincere intentions of responsible paternalistic leaders than with the self-serving bias of exploitative paternalistic ones. Both types of paternal leaders also believe in high institutional collectivism. Institutional collectivism" is defined as "the degree to which organizational and societal institutional practices encourage and reward collective distribution of resources and collective action", but the implementation and explanation of this term vary from organizational norms, institutional maturity, and the level of organizations. In emerging societies, where due to certain societal and economic constraints, people have to work under a prevailing leadership style, and workers experience different types of paternal leader.

In this paper, the researcher investigated the mechanisms through which personality traits of paternal leader affects employee job retention. We tested the model involving the personality traits of paternal leaders, virtual human resource practices, and employee performance in a service context. Our first finding is that a compassionate trait of a paternal leader (Potter 1969) was not positively associated with employee job retention n (Sheridan 1992), as reported by the education sector (H1). This confirms that a paternal leader can influence his traits (Nishii and Leroy 2022) with other factors that are important for employee retention in the organization. Our second hypothesis is also positive because it confirms that virtual human resources (Mondejar and Asio 2022) during the pandemic was an additional mechanism that links the compassionate trait of a paternal (Gilbert 2021) leader and work engagement activities (Christian et al. 2011). We then examined H3, which is rejected because normally, a paternal leader considers himself to be more experienced and has a greater right to decide business decisions, which can have a negative impact on the employees. H4 is accepted, and the reason is that it is a need of every organization that the leader must give priority to the system, which protects the employees of the organization from wrong decisions from direct leadership. H5 is also rejected because a paternal leader is not allowed to influence employee job retention (Gyamerah et al. 2022) when he or she is not fulfilling the requirement of a business environment, especially when the employee needs leader attention and support and the leader is not available for them and preferring there business goals and objectives instead of their employees' needs. The results indicated that the V-HRM (Mondejar and Asio 2022) consistency strengthens two relationships—empowerment trait—work engagement and empowerment trait of

paternal leaders (Gunasekara et al. 2022) and employee job retention (Sheridan 1992), respectively (H7).

In other words, employees are more likely to reciprocate excellent leader relationships and work passionately with higher job performance if they believe that Virtual HR (Ndidi et al. 2022) procedures in COVID-19 are highly consistent with one another. These findings show that there is a significant contingent relationship between leadership and outcome, and vice versa, with VHRM (Yussif 2022). Employees are better able to identify the management objectives driving HR practices (Yussif 2022) and more confidently interpret corporate goals when HRM messaging is consistent with one another. This encourages workers to respond to high leadership with acceptable attitudes and conduct by helping them connect their actions with the organization's aims and expectations. According to the co-variation (Nishii and Leroy 2022) principle of the attribution theory, employees will have a clear understanding of management expectations and associated rewards if they perceive their environment—in this case, HRM—as distinctive (it is visible) and consistent, and if other employees share their perception of HRM. As a result, they will be more likely to keep their jobs, and the effects of leadership on employee engagement (Wood et al. 2020) and job retention will be strengthened. After adjusting for HRM content, our findings showed that highly consistent HRM messages—as shown by within-respondent agreement—enabled workers to respond to leaders in ways that were consistent with the mission of the company. However, this article shed light on some studies that have found a non-significant relationship between paternal leadership and employee job retention (Yussif 2022) because our results also found a non-significant relationship between paternal leadership and employee job retention. Due to theCOVID-19 pandemic, a distinction was made between those who were the actual leader and those who were only supposed to present themselves as leaders, but in actuality, were not paternal leaders but deceptive people who never showed benevolence or resource (Elmes 2022) provider for their employees when people needed them duringCOVID-19. It is also a fact that inconsistent paternal policy leads to inconsistent HRM practices and subsequent confusion about the alignment of the employees' roles and the organization's objectives are likely to increase employee stress and dissatisfaction, leading to reduced performance.

In other words, without a defined aim, employees are more likely to experience stress, deliver subpar performance, and consider quitting. Following this logic, additional research may be beneficially focused on other management or HRM behaviors that can help employees' actions match company goals when looking at the results of paternal leader relationships.

### 6.1. Theoretical Implications

The COVID-19 pandemic has created a tremendous amount of uncertainty and challenges for business leaders and organizational members. In times of crisis, the leader becomes the public face for different stakeholders and plays a critical role in shaping an organization's direction. In particular, when faced with high uncertainty, middle managers pay close attention to their leaders to make sense of how their organizations will respond to and recover from this environmental shock and take action accordingly.

On a theoretical level, this research contributes to a new understanding in the area of paternal leadership on employee retention in the private colleges of Lisbon. First, the findings support the view that the paternal leader, as a resource provider, has significant impact on the employee retention of an institute. As a business grows, the supporting departments also play an important role, such as work engagement activities supported by the human resource department and leader secretariat.

The moderating effects lead to a more nuanced picture of how work engagement activities increase employee satisfaction and understand their reason for staying in the institute (Islam et al. 2022). We predicted that work engagement activities buffer the negative effects of leadership behavior during theCOVID-19 pandemic. As a result, employees who are more supported by their employers are more likely to have interpretative experiences that



will guide their longer stay in business. We found two groups in the educational institutions; the first group accepted that they were supportive of their organization, and a leader also provided them with opportunities in terms of understanding the work engagement that helps grow an organization, but COVID-19 changes the situation, the leader had a greater preference for financial policies and ignored achieving the long-term objectives of the firm. The second type of employee denied providing opportunities and desired work engagement activities which can help in thinking about their long stay in an organization. This is one of the first studies in the educational sector to employ a paternalistic leadership approach to describe the link between leadership and employees during COVID-19.

By identifying virtual human resource practices as a mediator in the relationship between paternal leadership and work engagement activities, we aimed to add to the study strategy under investigation. We examined the unexpected role that workplace engagement activities played in forecasting employee retention. By combining mediation and moderation, our model more effectively explains how leadership influences employee retention as well as who is most impacted by the choice of paternal leadership in terms of work engagement activities and employee retention. Our findings not only support and explain claims that the education industry is closely related to employee retention, but they also offer suggestions for reducing the negative effects of paternal leadership. Finally, in the context of Portugal, this study adds to academics' understanding of the impact of paternal leadership on employee work engagement activities and behavioral outcomes in terms of employee retention.

*6.2. Managerial Implications*

This study offers a number of managerial recommendations that can help CEOs, managers, educators, and politicians lessen the drawbacks of paternalistic leadership. The education sector will first benefit from the study's findings since they will help them better understands the variables that affect employee retention. Employee retention is crucial in making this field environmentally and socially acceptable. Employee retention, according to our findings, is an organizational phenomenon that is influenced by situational factors (e.g., paternal leadership and virtual human resource activities) through work engagement. According to the findings of this study, to attain long-term objectives, grow a business, and avoid financial losses in the long term, a deceptive attitude of leadership must be avoided. Second, people's responses to a paternal corporate culture are influenced by their earlier experiences; as a result, businesses may need to create a certain environment, reliable procedures, and achievement in order to support the culture of staff retention. Employees respond negatively to paternalistic leadership based on their personal experiences; therefore, executives and organizational leaders should establish moral standards. If they work in an ethically comfortable workplace with regular procedures and a formal complaint mechanism, employees may be inspired to do extra-role actions to improve organizational performance.

Third, unethical practices of the leader can be reduced by agency services provided by managers/administrators. It has been suggested that keeping the long-term objectives of the firm in view, continuity, the growth of an organization, and competition in the market, leaders need to understand what should be preferred and what is actually needed by an organization to grow. Finally, the pandemic was a difficult time where the leader should have to prove why he/she is blessed to be a paternal leader. The paternal leader should have to place priority on small financial losses instead of future losses. Finally, according to the findings of this study, virtual human resource practices and work engagement activities are significant predictors and influences of employee retention. Paternal leadership theory is based on employee well-being and assumes that a resource provider's benevolence and supportive side should be lighter.

*6.3. Limitations and Future Research*

As with all studies, this one has several limitations that must be taken into account before drawing conclusions. Due to the fact that the data were obtained from small- and medium-sized educational institutions, the conclusions of larger organizations may differ from ours. Second, the attitude, support, the contributions of paternal leader in their organizations before the pandemic is totally ignored. In order to better understand the connection between unethical leadership and employee retention, future research can examine the impact of demographics. Third, the results of this study cannot be extrapolated to other geographical regions because they are based solely on data collected from private colleges in a single province. Other regions of the world's research findings could be different from ours.

In conclusion, our research provides important messages to organizational leaders. The COVID-19 pandemic affected the world, and workers experienced different experiences. We saw that many paternal leaders behaved differently with respect to their titles. Most of the paternal style organizations did as their theory said according to them, but some who claimed to be very paternal within an organization were self-serving or deceptive, whose colors faded when put near the fire. Priority-based financial strategies were made to protect business leaders without keeping in mind the need for time. This affected the economy, and many people became jobless even though they did not receive any support from the government. This caused the economic cycle of Portugal to fall into a vicious cycle of poverty. This has also affected meeting the long-term objectives of organizations and the growth of the economy.

Now and onwards, the canvass of leadership styles does not look as attractive because people from emerging economies do not believe in any type of leader. It can also be concluded that a financial leader is the leader of himself only, regardless of what ever paternal style he adopts; he inspires people to serve himself, he wants to motivate the team to achieve his objectives, but it is very clear that team objectives and personal objectives are in opposing direction; they cannot be parallel. Business leaders in emerging economies need institutional maturity to achieve their dreams. Now that the COVID-19pandemic is over, the unprecedented effects of the pandemic are revealing new theories to guide people for neo-normal challenges.

In order to successfully mediate the relationship between unethical paternal leadership and employee retention, the current study sheds light on critical concerns about the unethical behavior of paternal leadership and employee retention. Work engagement activities are revealed as a key contingent factor. This study presents various techniques for reducing the detrimental consequences of unethical leadership and motivating staff to participate in work engagement activities, which has important organizational ramifications. The results of this study can also be utilized to launch future investigations into additional factors and the underlying processes that support employee retention.

Crises require leaders to take responsibility and do this visibly. By being visible and responsible, they are showing accountability and sharing risks with their followers, an important sign of solidarity with the many workers work place health and others who face personal risks during the pandemic.

**Author Contributions:** Author Contributions: Conceptualization, U.K.; methodology, M.R.; software, J.L.L.; validation, J.M.M.; formal analysis, R.M.D.; investigation, R.M.D.; resources, J.L.L.; writing—original draft preparation, U.K.; writing—review and editing, J.L.L.; visualization, M.R.; supervision, J.M.M.; funding acquisition, J.L.L. All authors have read and agreed to the published version of the manuscript.

**Funding:** This research was supported by Instituto Politécnico de Lisboa.

**Institutional Review Board Statement:** This study was approved by the Ethics Committee of Superior University. (approval date: 15 June 2022).

**Informed Consent Statement:** The study includes human subjects, and informed consent is applicable and taken at the time of fulfilling the survey form.

**Data Availability Statement:** The raw data supporting the conclusions of this article will be made available by the authors without undue reservation.

**Acknowledgments:** We thank Instituto Politécnico de Lisboa for providing funding for this study.

**Conflicts of Interest:** The authors declare no conflict of interest.

## Appendix A

**Table A1.** Factors loading.

| Variables | Items | Loadings |
|:---:|:---:|:---:|
| Alignment of Virtual Human Resource Practices with Work Engagement during COVID-19 | HRPwe1 | 0.627 |
| | Hrpwe2 | 0.661 |
| | hrpwe3 | 0.755 |
| | hrpwe4 | 0.824 |
| | hrpwe5 | 0.860 |
| | hrpwe6 | 0.655 |
| | hrpwe7 | 0.706 |
| | hrpwe8 | 0.611 |
| Compassionate trait of Leader | ctl1 | 0.760 |
| | ctl3 | 0.748 |
| | ctl4 | 0.836 |
| | ctl5 | 0.809 |
| Decision Trait of Leader | dtl1 | 0.794 |
| | dtl2 | 0.834 |
| | dtl4 | 0.794 |
| | dtl5 | 0.754 |
| Employee Job Retention | ejr1 | 0.826 |
| | ejr2 | 0.838 |
| | ejr3 | 0.882 |
| | ejr4 | 0.720 |
| | ejr5 | 0.719 |
| | ejr6 | 0.848 |
| Empowerment Trait of Leader | etl1 | 0.772 |
| | etl2 | 0.817 |
| | etl3 | 0.847 |
| | etl4 | 0.789 |
| | etl5 | 0.815 |
| | etl6 | 0.742 |
| Influential Trait of Leader | inflt1 | 0.701 |
| | inflt2 | 0.755 |
| | inflt3 | 0.806 |
| | inflt4 | 0.745 |
| | inflt5 | 0.790 |
| | inflt7 | 0.655 |
| | inflt8 | 0.690 |
| Organized Trait of Leader | otl1 | 0.876 |
| | otl2 | 0.855 |
| | otl3 | 0.810 |
| | otl4 | 0.798 |
| | otl5 | 0.781 |

**Appendix B  Consent Form and Questionnaire**

**Impact of Paternal Leadership on Employee Retention during COVID: Financial Crunch or Financial Gain**

**VOLUNTARY PARTICIPATION**

Your participation in this study is voluntary. It is up to you to decide whether to take part in this study. If you decide to take part in this study, you will be asked to sign a consent form. After you sign the consent form, you are still free to withdraw at any time and without giving a reason. Withdrawing from this study will not affect the relationship you have, if any, with the researcher. If you withdraw from the study before data collection is completed, your data will be returned to you or destroyed.

**CONSENT**

I have read, and I understand the provided information and have had the opportunity to ask questions. I understand that my participation is voluntary and that I am free to withdraw at any time, without giving a reason and without cost. I understand that I will be given a copy of this consent form. I voluntarily agree to take part in this study.

Participant's signature ______________________________ Date __________

Investigator's signature ______________________________ Date __________

| **Gender** | | | | |
|---|---|---|---|---|
| ☐ Male | ☐ Female | | | |

| **Age** | | | | |
|---|---|---|---|---|
| ☐ Less than 30 | ☐ 31–40 | ☐ 41–50 | ☐ Above 50 | |

| **Education Level** | | |
|---|---|---|
| ☐ Bachelor | ☐ Masters | ☐ Doctoral |

**What is your level in the organization?**

- Level 1—C-Suite/Executive/Registrar
- Level 2—Senior Leader/VP
- Level 3—Senior Manager
- Level 4—Team Lead
- Level 5—Non-leader Role

| **Work Experience** | | |
|---|---|---|
| ☐ Less than 10 years | ☐ 10 to 20 years | ☐ Above 20 years |

| | Strongly Disagree | Disagree | Neutral | Agree | Strongly Agree |
|---|---|---|---|---|---|
| **The influential Trait of my leader** | | | | | |
| My leader provides tasks to contribute in organization according to my skill? | | | | | |
| My leader helps me to make my long term carrier plan in the company? | | | | | |
| My leader supported me well for the skills which feel lacking? | | | | | |
| My leader provides me opportunity to achieve my personal development goals at my current position? | | | | | |
| My leader helps me to identify my personal goals and values? | | | | | |

My leader trains and guides me when I
feel having less ability at work?

**Decisive Trait of My leader**

My leader prioritizes which decisions
need to be made first.

My leader focuses on situation that
requires a decision and tries solve
that issue.

My leader allows others to be critical
about ideas and suggests solutions to
expand an idea.

My leader never takes the time to
actually consider what might go wrong.

My leader is a visionary and always
makes decisions with the future in mind.

**Organized Trait of Leader**

My leader is organized and dedicated to
his work

My leader is focused and attentive on
what is truly important.

The organized skill of my leader makes
him more productive and set an example
for team members as well.

My leader is able to juggle multiple tasks
and complete all of them efficiently and
effectively

My leader is flexible about work load,
task urgency and constructive criticism

**Empowerment Trait of leader**

My leader encourages me to
embrace change.

My leader recognizes the employee's
contribution to business

My leader reward me responsibly for in
workplace

My leader encourage me when I tried but
failed to achieve my tasks

My leader considered my mistakes as
learning opportunities instead
blaming me

My leader foster a culture of employee
engagement

My leader retain talent.

**Compassionate Trait of leader**

Do you believe your leader kind
behavior can impact on organizational
performance?

If my leader sees a colleague in need, he
go out to help

| |
|---|
| My leader remains unaffected when someone is unhappy |
| My leader prioritize an employee's wellbeing over a business outcome |
| My leader sacrifice a financially beneficial opportunity if it involves an ethically questionable business agreement |
| My leader sacrificed the short-term gain for longer-term objectives |
| **Alignment of Virtual Human Resource Practices with Work Engagement during COVID** |
| My job is aligned to new roles/responsibilities. |
| I am recruited, selected, trained, compensated and managed accordingly. |
| Due to technology increase in **COVID**, my job complexity increases, trained and compensated accordingly. |
| I am recognized and rewarded for voluntary contributions in COVID-19 and proactive work behaviors. |
| The institute culture emphasized pay-for-performance. |
| The culture of my institute focused on results-based incentives. |
| **Employee Job Retention in COVID** |
| Do you think that employee retention helps the development of organization? |
| Does your company pay more attention to incentives and perks offered to you? |
| You are encouraged to participate in training to improve your skills and Competencies? |
| Friendly relationships are encouraging in this organization? |
| Do you get the recognition that you deserve for your performance? |
| Would you like to plan your further career in this organization? |
| Do you think that the implementation of three R's (recognition, reward, respect) is practically applied in your organization and in future it will increase employee retention? |
| Does your management come forward to support when you are facing with critical situation in COVID? |
| Organization treating you in a respectful way? |

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
