# Peer review of "Impact of Paternal Leadership on Employee Retention during COVID-19: Financial Crunch or Financial Gain"

_socsci, doi:10.3390/socsci11100485_

Round 1
Reviewer 1 Report
Dear Author(s),
I must appreciate your work, still I have the following suggestions for its improvement:
1- In the introduction section, I would suggest talking about both negative and positive leadership and then focusing on paternalistic leadership. I, therefore, suggest the following papers on negative leadership:
Islam, T., Ahmad, S., Kaleem, A. and Mahmood, K. (2021), "Abusive supervision and knowledge sharing: moderating roles of Islamic work ethic and learning goal orientation", Management Decision, Vol. 59 No. 2, pp. 205-222.
Islam, T., Asif, A., Jamil, S. and Ali, H.F. (2022), "How abusive supervision affect knowledge hiding? The mediating role of employee silence and moderating role of psychological ownership", VINE Journal of Information and Knowledge Management Systems, Vol. ahead-of-print No. ahead-of-print. https://doi.org/10.1108/VJIKMS-11-2021-0274
Ahmad, S., Islam, T., Sohal, A.S., Wolfram Cox, J. and Kaleem, A. (2021), "Managing bullying in the workplace: a model of servant leadership, employee resilience and proactive personality", Personnel Review, Vol. 50 No. 7/8, pp. 1613-1631.
- Then there are few important papers on the used variables that have been ignored. For example:
Chaudhary, A., Islam, T., Ali, H.F. and Jamil, S. (2021), "Can paternalistic leaders enhance knowledge sharing? The roles of organizational commitment and Islamic work ethics", Global Knowledge, Memory and Communication, Vol. ahead-of-print No. ahead-of-print. https://doi.org/10.1108/GKMC-06-2021-0109
Islam, T., Ahmad, R., Ahmed, I. and Ahmer, Z. (2019), "Police work-family nexus, work engagement and turnover intention: Moderating role of person-job-fit", Policing: An International Journal, Vol. 42 No. 5, pp. 739-750.
2- Hypotheses of the study are acceptable, although detailed information can strengthen the same.
3- The methodology of the study is in detail and well discussed, just mention why preferred Smart-PLS SEM.
4- Results:
- in the assessment of the measurement model, I suggest adding model fit indices as well. A separate heading of factor loading can be skipped, and all the values can be presented in the appendix.
- In the interpretation of regression analysis, I suggest going through the following paper, and reporting the results accordingly as it is a more professional way of doing the same:
Islam, T., Ahmad, S. & Ahmed, I. Linking environment specific servant leadership with organizational environmental citizenship behavior: the roles of CSR and attachment anxiety. Rev Manag Sci (2022). https://doi.org/10.1007/s11846-022-00547-3
5- Discussion, theoretical and practical implications are there; but I guess, the heading for the conclusion can be skipped as everything has already been discussed in the discussion and limitation sections.
6- The manuscript is easy to read.
Author Response
Dear Sir,
Please see the attachment
Reviewer 2 Report
Dear Authors,
I read your manuscript entitled, "Impact of Paternal Leadership on Employee Retention during COVID: Financial Crunch or Financial Gain". The topic of this study is very interesting. However, the following concerns need to be considered to make this manuscript more understandable.
Literature
1. The last paragraph of the Literature section (from Line 327 to Line 341) must be removed from the manuscript. Please be careful when using the journal template.
2. Theoretical model displayed in Figure 1 should be placed at the end of the Literature section, immediately below the Hypotheses justification
Research Methodology
3. I would appreciate it if the authors provided a certain explanation of using the positivist research paradigm.
4. In Line 348 and Line 349, the authors mentioned, "[…] "In order to address the objectivity of research issues, as recommended by experts, the study uses a positivist research paradigm". Please justify your statements according to previous studies.
5. In the sampling subsection, the authors indicated that they calculated the sample size using the Rao-soft software. Please provide more details about the sample size calculation.
6. Concerning data collection methods, a complete description of how data were acquired (population, sample size, operationalization of constructs, data collection period, etc.) should be included.
7. The authors must include descriptive statistics of respondents who participated in the study.
8. In the sampling subsection, it was shocking that the authors indicated that the "Complete questionnaires were not included in the analysis". Do you mean incomplete questionnaires?
9. In the instrument subsection, the authors should provide more details about the sample used in the pilot research. The authors only report the number of participants in the pilot study. The author should explain more regarding the pilot study step.
10. The PLS-SEM was used in this study, it is better to provide the reason for employing this approach. In addition, it is crucial to provide the reason for using Smart-PLS-3 software, and do not forget to cite Ringle, Wende & Becker (2015).
11. Referring to your statement in lines 397 and 398, the authors mentioned, "Several researchers, including Hair et al. (2019), have concluded that factor loadings are the most important indicator of validity. Items with values greater than 0.50 are kept in the database". However, by referring to Hair et al. (2019), it is clear that they recommended values of factor loadings above 0.708 to keep items. More specifically, in the step of outer loading relevance testing, an item with an outer loading value >0.4 and < 0.7 can be retained only if its removal does not increase the values of AVE and CR.
12. The author should explain more regarding the results of reliability and convergent validity presented in Table 1. In addition, I recommend reconsidering the title of table 1. I propose using this title: reliability and convergent validity results.
13. The factor loadings subsection should be placed before the discriminant validity results.
14. The readers can benefit from the questionnaires used for each construct, please include them in Table 3. As well, instead of keeping the table empty without any loading value for the excluded items (i.e. items ctl2, dtl3, etl7,), it may be more suitable to specify the loading value of these items by adding the indication "eliminated from the scale".
15. Figure 2 is not clear and does not show factor loadings values. Please check the figure and include the appropriate one.
Assessment of Measurement Model
16. In the discriminant validity subsection, the authors indicated that "Henseler and Ringle asserted that the HTMT referred as, hetero- trait-to-monotrait ratio must be smaller than 0.85 […]. Because all of the values in table2 are less than 0.85, there is no concern with discriminant validity". It is good to employ the Heterotrait-Monotrait Ratio as an alternative to the Fornell-Larcker Criterion when assessing outer models' discriminant validity. Actually, Table 2 does not show the HTMT ratio. Therefore, please include a table showing discriminant validity results using the HTMT ratio.
17. I would also invite the authors to introduce other essential criteria for checking the structural model, including the endogenous latent variables coefficient of determination (R2), the effect size (f2), the predictive relevance (Q2), and the model goodness-of-fit (GoF).
Minor issues:
18. A number of sentences need to be rephrased because they are copied from Rafiq et al. (2020). https://doi.org/10.3390/su12041365. For instance
Lines 343 and 344 "This segment of the research contains the methodology, the instrument, data collection tool, and sampling techniques used in the study to investigate the relationship".
Lines 369 and 370 "the varimax rotation from principal component analysis and confirmatory factor analysis were"
Lines 375 and 378 " The sample size was calculated through Rao-soft software, which suggests the sufficiency of respondents as Equations (1)–(3) …"
19. I can note that reference citations in the text do not appear in the references section. For instance, Hair et al. (2019); Hair et al. (2016) do not appear in the reference list. Please ensure that every reference cited in the text is also present in the reference list (and vice versa).
20. Some minor English editing is needed to correct the various grammatical errors present in the manuscript. I strongly recommend professional proofreading for the entire manuscript.
I hope these comments are useful for further steps.
All the best
Author Response
Dear Sir,
Please see the attachment.

Reviewer 3 Report
The proposed research is intriguing. The authors used a fitting measurement model and suitable regression analysis. The analysis is well done, and the presentation of the results is clear and coherent.
The research does not compare the paternal contributions of participants before the pandemic, which would have been valuable to evaluate and compare the impact of paternal contributions during Covid 19.
Issues are with the presented (or the lack of such) data. Questionnaires dedicated to identifying personality types or leadership styles are controversially discussed in academia. To ensure the transparency of the research as well as evaluate the quality of the study, the presentation of the question set is necessary. The authors did not provide the question set. Without these, an analysis of the quality of the research is not possible. Personality trait questionnaires bear problems of participant biases. Too often, participants would answer in such a way that influences the outcome (to leave a good impression).
The number of hypotheses is too high, which led to a very brief analysis of each hypothesis. It might have been better to reduce the hypothesis for this manuscript.
As the primary research included human beings, information about informed consent, ethics approval, etc., is missing.
Author Response
Dear Sir,
Please see the attachment

Round 2
Reviewer 2 Report
The authors have done a very good job in addressing the reviewers' outstanding issues. This version has improved significantly, which only requires some minor revisions as follows:
(1). The authors must include descriptive statistics of respondents who participated in the study.
(2). In the instrument subsection, the authors should provide more details about the sample used in the pilot research. The authors only report the number of participants in the pilot study.
(3). Please include a table showing discriminant validity results using the HTMT ratio
(4). Please verify Table 4, the head of the table specifying the used indicators is missing.
All the best,
Author Response
See the Uploaded file, please.

Reviewer 3 Report
Dear authors,
Congratulations on your research. The manuscript in its current version meets the requirements for publication. Some of your questions to identify the leadership traits could have been improved, but the effect on the data is minimal.
